# Mechanisms of COVID-19 Associated Pulmonary Thrombosis: A Narrative Review

**DOI:** 10.3390/biomedicines11030929

**Published:** 2023-03-16

**Authors:** Cristian-Mihail Niculae, Adriana Hristea, Ruxandra Moroti

**Affiliations:** 1Infectious Diseases Department, Faculty of Medicine, University of Medicine and Pharmacy “Carol Davila”, 37 Dionisie Lupu Street, 020021 Bucharest, Romania; adriana.hristea@umfcd.ro (A.H.); ruxandra.moroti@umfcd.ro (R.M.); 2National Institute for Infectious Diseases “Prof. Dr. Matei Bals”, 1 Calistrat Grozovici Street, 021105 Bucharest, Romania

**Keywords:** COVID-19, SARS-CoV-2 infection, pulmonary in situ thrombosis, embolism, immunothrombosis, inflammation, coagulopathy

## Abstract

COVID-19, the infectious disease caused by severe acute respiratory syndrome coronavirus-2 (SARS-CoV-2), is frequently associated with pulmonary thrombotic events, especially in hospitalized patients. Severe SARS-CoV-2 infection is characterized by a proinflammatory state and an associated disbalance in hemostasis. Immune pathology analysis supports the inflammatory nature of pulmonary arterial thrombi composed of white blood cells, especially neutrophils, CD_3_^+^ and CD_20_^+^ lymphocytes, fibrin, red blood cells, and platelets. Immune cells, cytokines, chemokines, and the complement system are key drivers of immunothrombosis, as they induce the damage of endothelial cells and initiate proinflammatory and procoagulant positive feedback loops. Neutrophil extracellular traps induced by COVID-19-associated “cytokine storm”, platelets, red blood cells, and coagulation pathways close the inflammation–endotheliopathy–thrombosis axis, contributing to SARS-CoV-2-associated pulmonary thrombotic events. The hypothesis of immunothrombosis is also supported by the minor role of venous thromboembolism with chest CT imaging data showing peripheral blood clots associated with inflammatory lesions and the high incidence of thrombotic events despite routine thromboprophylaxis. Understanding the complex mechanisms behind COVID-19-induced pulmonary thrombosis will lead to future combination therapies for hospitalized patients with severe disease that would target the crossroads of inflammatory and coagulation pathways.

## 1. Introduction

COVID-19, the infectious disease caused by severe acute respiratory syndrome coronavirus-2 (SARS-CoV-2), is frequently associated with micro- and macrovascular thrombotic events, which are associated with disease severity and worse clinical outcomes [1,2,3]. Severe hypercoagulability mainly develops in the lungs of COVID-19 patients [4], and the most common vascular thrombotic complication involves the pulmonary arteries [1,5,6]. The incidence of pulmonary artery thrombosis in patients with COVID-19 pneumonia varies between 18 and 57%, with a pooled determined incidence of 30.2 [7]. Post mortem lungs from patients with SARS-CoV-2 infection showed severe coagulation abnormalities, especially fibrin- and platelet-rich thrombi, not observed in non-COVID-19 autopsy controls [4]. The “cytokine storm” and its associated diffuse endothelial dysfunction, common in severe SARS-CoV-2 infection, could also increase the risk of developing nonrespiratory complications, such as neurological involvement, preeclampsia, and orchitis-like syndromes [8,9,10]. COVID-19-associated thrombotic complications are secondary to a synergistic interplay of endotheliopathy, coagulation pathways, platelet dysfunction, and detrimental immune-mediated thrombosis [11]. However, it is not clear which of the pathobiological mechanisms, conventional risk factors, and venous thromboembolic disease, in situ immunothrombosis, or additional thrombotic mechanisms contributes more to these COVID-19-associated pulmonary thrombotic events [2,12].

In this narrative review, we aimed to explore the pathophysiological pathways supporting the potential mechanisms responsible for COVID-19-associated pulmonary thrombosis, which could have implications for diagnostic and management strategies. We conducted free searches in the PubMed database for publications reporting on patients with COVID-19 and pulmonary thrombosis and/or associated mechanisms. Clinical, pathobiological, and imaging data were extracted. In papers where the mechanism of thrombosis (embolism or local clot formation) could not be identified, we used the term pulmonary thrombosis covering both pulmonary embolism and in situ thrombosis. We searched only papers in English, using the following terms in various combinations: “SARS-CoV-2”, “COVID19”, “novel coronavirus”, “venous thromboembolism”, “thrombosis”, “pulmonary embolism”, “pulmonary thrombosis”, “immunothrombosis”, and “in situ thrombosis”. We restricted our search to papers with full text or abstracts written in English and no other restrictions were applied. We considered relevant for this narrative review data from original human studies, preclinical/animal studies, and case reports as well as previously published reviews.

## 2. COVID-19-Associated Pulmonary Thrombosis Is an In Situ Immunothrombosis

Pulmonary thrombosis in situ is a pathological condition nonrelated to embolism from deep vein thrombosis (DVT) in the lower extremities [13]. Nonpulmonary in situ thrombosis in the right ventricle was also described in COVID-19 patients [14,15]. COVID-19-associated pulmonary intravascular coagulopathy is a complex disease “orchestrated” by a severe and dysregulated proinflammatory response that can lead to immunothrombosis [2,16]. The prothrombotic state in patients with COVID-19 is reminiscent of this immunothrombosis process, a result of the crosstalk between the immune and hemostatic systems and characterized by the production of microthrombi in small capillaries, in which endothelial cells (ECs) adopt a proadhesive phenotype in contact with SARS-CoV-2 [17,18,19]. Pathological studies showed multiple microthrombi in pulmonary capillaries and larger primary thrombi in arterioles considered to be primary in nature [15]. Autopsy lung samples from patients with COVID-19 showed important circulatory changes with inflammation-dependent intravascular thrombosis, direct pathological evidence for immunothrombosis, which were not found in other organs such as heart, brain, and kidneys [15]. There is a concept of a local lung-associated coagulation system, the “bronchoalveolar hemostasis” and the formation of blood clots in the microvasculature of the lungs could be a part of the host immune defense against SARS-CoV-2 [20,21]. A normal host alveolar capillary associated immune defense system is presented in Figure 1.

In patients with in situ pulmonary thrombosis, in-depth immune pathology analysis by immunohistochemistry supports the inflammatory nature of arterial thrombi composed of white blood cells, especially neutrophils, CD_3_^+^ and CD_20_^+^ lymphocytes, fibrin, red blood cells, and platelets, but not megakaryocytes [16]. Regional thrombosis of the lung microvasculature also showed compressed deformed red blood cells (RBCs), including polyhedrocytes and different morphological types of fibrin structures coated with sparse spherical microparticles, which could comprise virions or cellular ectosomes [15]. Different phenotypes of in situ thrombosis may exist and need different intensity of anticoagulant therapy [22].

Severe COVID-19 is characterized by a proinflammatory state and an associated disbalance in hemostasis, which starts with the disruption of the alveolar epithelium, and involves intrinsic and extrinsic coagulation pathways, neutrophil extracellular traps (NETs) activation and release (NETosis), and impaired fibrinolysis secondary to high plasminogen activator inhibitor 1 (PAI-1) levels [18]. Acute respiratory failure in patients with severe COVID-19 is associated with diffuse alveolar damage, perialveolar microangiopathy, and obstructive neoangiogenesis [18,23,24,25,26,27,28,29,30]. In situ pulmonary thrombosis may appear in COVID-19 pneumonia patients, with peripheral distribution, either within the consolidation lesions of the infected lungs (due to active local inflammation), or in nonconsolidation areas (due to hypercoagulability caused by systemic inflammation) [22]. Systemic inflammatory markers such as C-reactive protein (CRP) have higher values in patients with COVID-19-associated pulmonary artery thrombosis compared with those without pulmonary thrombi [31,32,33,34]. Inflammatory-mediated thrombosis is also supported by the similarity in terms of comorbidities and other classical risk factors for venous thromboembolic disease in patients with and without COVID-19-associated pulmonary thrombosis [31,32]. Elevated serum levels of CRP are associated with thrombotic disease and mortality in COVID-19 patients [35,36,37,38]. Moreover, CRP is an important link between inflammation and thrombosis, as it can activate the complement cascade, induce platelet adhesion to ECs, stimulate tissue factor (TF) expression by blood monocytes, and alter the fibrinolytic balance of ECs [39]. Ferritin, another inflammatory biomarker, is also an independent predictive factor for both venous thrombotic events, especially pulmonary thrombosis, and also mortality, in COVID-19 patients [6,40,41]. Ferritin induces mitochondrial dysfunction in platelets, and this could also contribute to inflammation and thrombosis [2].

## 3. Biological Mechanisms for In Situ Pulmonary Immunothrombosis

### 3.1. Inflammatory Pathways

#### 3.1.1. Macrophages (AMφs), Monocytes, and T Cells

Pathogen-associated molecular patterns (PAMPs) and endogenous-damage-associated molecular patterns (DAMPs) release through cellular injury, and, when recognized by specialized receptors such as toll-like receptors (TLRs) and C-type lectin receptors (CLRs), trigger intracellular signaling cascades and produce various inflammatory cytokines and chemokines that modulate coagulation abnormalities, especially those mediated by the nuclear factor kappa-light-chain-enhancer of activated B cells (NF-κB) signaling pathway [42]. The SARS-CoV-2 spike (S) protein is a TLR-4 ligand and an NF-κB pathway activator in monocytes as well as in macrophages, and its RNA can activate TLR-3 and TLR-7 cellular signaling pathways [43]. The synthesis of interferons (IFNs): IFN-α, IFN-β; interleukins (ILs): IL-1α, IL-1β, IL-2, IL-6, IL-8; and tumor necrosis factor-alpha (TNF-α) can contribute to inflammation and thrombosis through multiple pathways [43]. Comparing inflamed with normal alveoli areas from the same alveolar biopsies of COVID-19 lungs, in inflamed alveoli, upregulated genes were enriched for innate immune and inflammatory pathways, including neutrophil degranulation and IFN-γ and interleukin signaling [44]. As SARS-CoV-2 replicates in the type II alveolar epithelial cells and interacts with pulmonary microvasculature at the pneumocyte–capillary interface, it locally activates the innate immunity leading to an exaggerated release of proinflammatory cytokines in severe COVID-19 cases, known as the “cytokine storm” [45]. Macrophages, monocytes, CD_4_^+^ T cells, and CD_8_^+^ T cells are increased and activated in COVID-19 patient lungs [4]. A cell census from the COVID-19 lung autopsies also showed increased levels of dendritic cells (DCs) and natural killer (NK) cells [44]. In severe COVID-19, in the setting of inflammation, tissue resident AMφs in the bronchoalveolar space are diminished in number, a phenomenon described as the macrophage “disappearance reaction” [46]. Infected tissue resident AMφs express higher levels of proinflammatory chemokines, which leads to an excessive monocyte recruitment through a positive feedback loop [46]. Monocytes, in addition to their role against various pathogens, promote the activation of the extrinsic coagulation pathway during inflammatory states and could be considered a bridge from inflammation to thrombosis [47]. A strong association of platelet–monocyte interaction with monocyte inflammatory activation and immune system dysfunction is seen in critically ill patients, contributing to thromboinflammation in COVID-19 [48]. Moreover, the axis NETs–CD_14_^+^/CD_16_^+^ inflammatory monocytes triggers an excessive amount of IL-6 and other proinflammatory molecules via a positive feedback mechanism, also contributing to the “cytokine storm” [45].

#### 3.1.2. The NETs–Thrombosis Axis in COVID-19

SARS-CoV-2-induced “cytokine storm” primes neutrophils to release NETs, a distinct mechanism of innate immune response composed of web-like structures containing DNA filaments coated with histones and granule proteins, that can entrap and eliminate various pathogens [49]. The disbalance of vascular microenvironmental homeostasis and aberrant NETs formation leads to pulmonary thrombosis, and it may augment the SARS-CoV-2-induced “cytokine storm” and macrophage activation syndrome in patients with severe disease [50,51]. NETs, together with platelets, RBCs, and procoagulant molecules (von Willebrand factor (vWF), fibronectin, fibrinogen, FXII, and TF), contribute to SARS-CoV-2-associated thrombotic events [11,19]. Pulmonary autopsies in patients with COVID-19 confirmed NETs-containing microthrombi with neutrophil–platelet infiltration [52]. NETs are responsible for the interplay between inflammation and thrombosis in the lungs of COVID-19 patients, promoting tissue injury and secondary immunothrombosis [49,53]. High levels of markers of NETs are found in the plasma of patients with severe COVID-19 [19,52,54,55]. NETs are implicated in SARS-CoV-2-associated acute respiratory distress syndrome (ARDS) pathogenesis, being present in high levels in the lower respiratory tract samples of these patients but also in lung tissue from deceased patients with COVID-19 [56]. TLR2 and C-type lectin domain family 5 member A are critical in the induction of NETosis in lung tissues by the SARS-CoV-2 S protein, leading to COVID-19-associated pulmonary thromboinflammation [57].

#### 3.1.3. Mast Cells (MCs), Cytokines, and Chemokines

MCs are specialized subendothelial innate sentinel cells characterized by antigen processing capabilities and being equipped with TLRs and other receptors involved in different inflammatory pathways, such as FcεRI, MAS-related G protein-coupled receptor-X2 (MRGPRX2), IgG receptors, Fc-gamma type 2 receptor A (FcγRIIA), dectin-1, IL-10 receptor, and substance-P and complement receptors [58,59,60,61,62,63]. Viruses can activate MCs directly or by viral particles to release preformed inflammatory mediators, vasoactive autocoids such as histamine and catalytically active MC-specific proteases, including β-tryptase, chymase, Granzyme B, carboxypeptidase-3, and β-hexosaminidase [62,63]. MCs also produce de novo lipid mediators such as prostaglandins (PGD2, PDE2), leukotrienes (LTB4, LTC4, LTD4), platelet-activating factor (PAF), cytokines (e.g., TNF, IL-6, IL-4, IL-5, IL-1β, IL-10, IL-13), and chemokines (e.g., CCL1, CCL2, CXCL1, CXCL8) [58,59,60,62], many of which are now known to be associated with the “cytokine storm” observed in severe COVID-19.

Clusters of degranulating MCs expressing chymase and tryptase are seen in the lung areas with hemorrhagic phenomena, and it could be connected to local histamine production, stored endogenously within the secretory granules of MCs, and released into the vessels after cell stimulation [64,65]. Resident phagocyte alveolar macrophages activated by SARS-CoV-2-TLR interaction produce IL-1 which further stimulates MCs to produce IL-6 [65]. IL-1 causes microthrombi and inflammation by promoting EC–leukocyte adhesion, endothelial dysfunction, and thromboxane A2, B2 (TxA2, TxB2), and TNF-α production [66].

Accumulation of MCs in the lungs can cause inflammation and thrombosis. In healthy lung tissue, MCs express low levels of critical cell-entry facilitators for SARS-CoV-2, such as angiotensin converting enzyme (ACE)-2 and its serine protease for S protein priming, transmembrane serine protease 2 (TMPRSS2) [67,68]. When analyzing mRNA expression of autoimmune-related genes in the lung tissue, MCs were the only type of all studied cells (MCs, neutrophils, macrophages, exhausted CD_8_^+^, CD_8_^+^ T cells, dendritic cells, cytotoxic cells, B-cells, CD_45_ cells and T-cells) that were highly expressed in patients with SARS-CoV-2 pneumonia compared to influenza [69]. A total of 43 genes were significantly differentially expressed between these patients, including chemokine C-X-C motif ligand 7 (CXCL7), which is present in platelets after thrombus formation followed by neutrophil attraction [69]. Microthrombi, present in more than half of the cases, as well as thrombi in small or large arteries, were more often seen in COVID-19 than influenza [69]. In SARS-CoV-2 infected patients, even without the replication of the coronavirus inside MCs, the interaction S-receptor binding domain (RBD)-ACE2 induces rapid MC degranulation/activation [70]. Proteases released from MCs further enhance cell viral entry, as MC-derived chymase interacts with SARS-CoV-2 S protein [71]. The MC degranulation degree rather than their total number is associated with the prothrombotic phenotype typical of COVID-19 [72]. Under IL-4 expression, early recruiting of CD_117_^+^ MC progenitors in the alveolar septa will lead to MC proliferation/differentiation, and once activated, they will orchestrate the crosstalk between proinflammatory and procoagulative networks, such as the complement and the plasma kallikrein-kinin system [73]. Some of the cytokines associated with SARS-CoV-2 “cytokine release syndrome”, IL-2, IL-4, IL-6, IL-8, IL-1β, IL-13, IL-12, TNF-α, and IL-7, especially IL-8 and TNF-α, further contribute to MC chemotaxis [74,75,76]. MC activation with subsequent degranulation in the respiratory tract submucosa will release high levels of IL-1, IL-6, and TNF-α, but also other proteins such as matrix metalloproteinase 9 (MMP-9), PAF, substance P, transforming growth factor beta (TGF-β), TXB2, and vascular endothelial growth factor (VEGF), which will contribute to the pathogenesis as proinflammatory and prothrombotic molecular factors [77,78]. MCs and their associated proteases, chymase (CMA1), carboxypeptidase A3, and tryptase beta 2, also modulate indirectly the systemic thrombo–inflammatory response in patients with SARS-CoV-2 pneumonia, as they induce thrombosis through activation of clotting factors and platelets, and this may affect the relatively high incidence of pulmonary thrombotic events in COVID-19 [69,72].

COVID-19-associated coagulopathy with microvascular thrombosis is secondary to inflammation and endotheliopathy that are “orchestrated” by IL-6, a pleiotropic proinflammatory cytokine [79]. The severity of SARS-CoV-2 infection is associated with increased blood levels but also with IL-6 expression on lung tissue [79,80]. IL-6 has a proinflammatory role in vascular endothelial cells, and it favors hypercoagulation by interfering with the normal anticoagulant and profibrinolytic properties of ECs [81]. IL-1, another key proinflammatory cytokine, had the best correlation with COVID-19-associated thrombotic events compared to IL-6 and TNF-α in one study [82]. IL-1-mediated inflammation in COVID-19-associated acute lung injury (ALI) follows the biological path of NLRP3 inflammasome and caspase-1 activation, leading to production of major innate immune mediators, IL-1β and IL-18 [81]. A bidirectional relationship also exists between IL-1-mediated inflammation and coagulation [81]. IL-1 maintains thrombosis by increasing the time of clot lysis, upregulation of TF expression, and activation of the endothelium via the IL-1β pathway, also promoting the recruitment of leukocytes [81]. In one study, in patients with COVID-19-associated thrombotic events, no statistical significance was recorded between serum levels of inflammatory markers (CRP, ferritin), age, and other clinical characteristics, except for IL-1β and soluble P-selectin [83].

#### 3.1.4. Complement Pathways

The complement system is a key mediator of the innate immune response and inflammation [84]. Lectin and alternative pathways mediate complement activation by SARS-CoV-2 S and N proteins [85,86,87], which will lead to EC and MC degranulation, enhanced phagocytic activity of neutrophils and monocytes, and “orchestration” of a proinflammatory environment [84]. Cleavage components C3a and C5a are potent inflammatory molecules capable of inducing the release of proinflammatory cytokines as part of the COVID-19-associated “cytokine storm” [87]. Plasmin-mediated C5 activation is also involved in thrombosis through multiple mechanisms [88]. The intricate network between the complement system and the coagulation cascade could sustain an enhanced rate of COVID-19-associated coagulopathy [89]. Severe COVID-19 with respiratory failure is characterized by a diffuse lung microvascular injury mediated by complement activation indicated by both the presence of significant serum levels and deposits in lung microvasculature of terminal complement components C5b-9 (membrane attack complex), C3b, and C4d, [87,90,91,92]. Alternative Pathway Activator factor D and Lectin Pathway Activator Mannan-Binding Lectin-Associated Serine Protease 2 were also markedly increased in lungs from patients with COVID-19 [87,91]. Colocalization of S glycoproteins with C5b-9/C4d in the interalveolar septa in autopsy lung samples from patients with COVID-19 suggest a direct, local complement activation by SARS-CoV-2 [90]. Moreover, this cross talk between complement, immune, and coagulation systems amplifies the process of COVID-19-associated coagulopathty by a positive feedback loop [19].

Immunothrombosis is an interplay between systemic and lung inflammatory pathways and coagulation/fibrinolysis systems (Figure 2).

### 3.2. Coagulation Pathways

#### 3.2.1. ECs and Platelets

COVID-19 is an endothelial disease, triggering the “cytokine storm” and inflammation, oxidative stress, and coagulopathy [93]. SARS-CoV-2 S protein–ACE-2 interaction activates the endothelium, which will lead to stimulation of immune responses and EC damage, activation of coagulation, reduction of fibrinolysis, and platelet adhesion and aggregation [94]. Damaged ECs leads to exposure of the prothrombogenic basal membrane, upregulation of TF, and release of coagulation factor VIII, vWF, and P-selectin from Weibel-Palade bodies (WPBs) [19]. Markers of ECs and platelet activation are significantly elevated especially in severe and critical SARS-CoV-2 infection, leading to diffuse endotheliitis and impaired microcirculatory function with an associated procoagulant state [95,96].

COVID-19 is also a disease of platelets pathology [97]. Platelets have the ability to interact with viruses, bacterial pathogens, and *Plasmodium* parasites [98]. SARS-CoV-2 S protein can interact with integrin α5β1, expressed on platelets [99]. Platelets are able to “orchestrate” monocyte responses to inflammatory activation, inflammatory cytokines secretion, and TF expression in COVID-19 patients [48]. Thrombocytopathy, as well as platelet-associated abnormal hyper-reactivity phenotype in COVID-19, especially in the lungs, contributes to severe hypercoagulability and inflammation [4,100]. Key proinflammatory interleukins in COVID-19, IL-6, and IL-1, also interact with the platelet–thrombosis pathway. IL-6 induces platelet reactivity and high fibrinogen linking inflammation and thrombosis [29]. Platelet numbers and their degranulation activity are associated with IL-1β plasma concentration [101]. In COVID-19 patients, blood platelets showed upregulated receptors, contributing to aggregation and activation [4]. Upon activation, granules containing proteins with hemostatic functions are transferred to platelet membrane and released to extracellular space to further promote platelet adhesion and activation [102]. The lung autopsies of COVID-19 patients showed upregulated platelet expression of PF4 (CXCL4), which promotes blood coagulation and activates the NF-κB signaling pathway in ECs, acting as a potential agent of both thrombosis and inflammation [4,103]. Anti-PF4 antibodies may be involved in the pathophysiology of severe clinical complications of COVID-19 [104]. Significant higher serum levels of other platelet activation markers involved in cell adhesion such as LAMP-3, as well as transmembrane proteins from the GPIIb/GPIIIa complex, vWF receptor units (GPIbα, GPIX), CD9, and CD40 (TNFRSF5, a transmembrane TNF superfamily receptor) were recorded in patients with SARS-CoV-2 infection [105]. However, the lung autopsies of COVID-19 patients showed that platelet expression of some receptors such as CD40L, CD42b (vWF receptor units, GPIbα), and CXCR4 were comparable to the control group [4]. CD40L and CXCR4 are involved in cell signaling in innate, adaptive immunity and thrombosis, mediating the acute lung injury (ALI) [102,106,107,108]. According to the same study, surface adhesion molecule expression such as ICAM-1, VCAM-1, and E-selectin was also reduced in lung ECs, suggesting that other mechanisms would be involved in the pulmonary infiltration of immune cells [4].

#### 3.2.2. vWF

vWF is a multimeric glycoprotein procoagulant molecule synthesized by ECs and stored in lysosome-related organelles such as WPBs [109,110]. SARS-CoV-2-mediated EC activation and damage in the pulmonary vascular bed by proinflammatory cytokines, complement activation, NETosis, and hypoxia increase local *vWF* release from WPBs [11]. ECs in the lungs of COVID-19 patients exhibit an increase in both hypoxia-inducible factor 1-alpha (HIF-1α) and glucose transporter protein type 1 (GLUT1) expression, confirming local hypoxic stress as a potential trigger for this pathogenic process [4]. The lung autopsies of COVID-19 patients suggest increased vWF expression not only from ECs, but also by activated platelets, as it was detected in the outer layer of the thrombus and CD31-expressing vascular endothelium tissue samples [4]. vWF plays a critical role in thromboinflammation, with NFκB2 mediated vWF transcription as a potential direct biological link between immunity and thrombosis [11,111,112]. By using the TLR2-dependent activation of the NF-κB pathway in a MyD88-dependent manner, SARS-CoV-2 S protein induces inflammation and promotes vWF transcription, causing in situ thrombosis, as vWF binds to platelets, neutrophils, and monocytes [113,114,115]. ADAMTS13 functional metalloprotease cleaves newly released highly active ultra large vWF (UL-vWF) multimers into smaller, less thrombogenic and inflammatory fragments [11,109,111]. In COVID-19, the ADAMTS13/vWF ratio decreases, leading to an excess of overactive UL-vWF multimers as a key driver of microthromboses in the pulmonary vasculature and SARS-CoV-2-associated ARDS [11,95,111,116,117,118,119]. Plasma exchange could be a therapeutic option for critically ill COVID-19 patients in order to restore the disbalance in the ADAMTS13-vWF axis [116,117].

#### 3.2.3. Thrombomodulin and P-Selectin

Pulmonary vascular thrombosis is prevented by the EC monolayer that expresses surface nonsoluble thrombomodulin (nsTM), a type I transmembrane glycoprotein with important anticoagulant activity, secondary to a protein C-mediated pathway and a direct modulation of thrombin’s procoagulant effects [115,120]. Thrombomodulin also has cytoprotective and anti-inflammatory properties, as it blocks the TLR-4-specific ligands’ axis, inhibits nuclear translocation of NF-κB, and induces direct mediated anti-inflammatory signals mediated by the endothelial protein C receptor (EPCR)-protease-activated receptor 1 (PAR1) system [120]. A dysfunctional prothrombotic phenotype of vascular ECs in the lungs of COVID-19 patients is characterized by a decreased expression of nsTM and other anticoagulants such as EPCR [4]. Decreased nsTM expression in ECs is associated with increased immune cell infiltration in the lungs of patients with SARS-CoV-2 infection [4]. Moreover, dysregulated proinflammatory cytokine generation by SARS-CoV-2 in the pulmonary microvasculature releases soluble thrombomodulin (sTM) from EC surfaces that will further promote a procoagulant and proinflammatory local milieu [115]. Administration of a human recombinant thrombomodulin in sepsis-induced coagulopathy could have some favorable effects in terms of patient’s outcome [121]. Endothelial thrombomodulin expression could also be a potential therapeutic target for COVID-19-related immunothrombosis [4].

P-selectin, another biomarker of endothelial degranulation, blocks the initial attachment and rolling of platelets and leukocytes to inflammatory regions [17]. In COVID-19 patients, P-selectin could be a biomarker of severe disease and associated venous thrombosis [17,83,119,122,123,124]. P-selectin might also be detrimental in COVID-19 as it promotes NETosis through binding to P-selectin glycoprotein ligand-1 (PSGL-1), leading to the development of immunothrombosis and ALI/ARDS [51,125]. P-selectin could also be a marker for platelet and endothelial activation weeks, even up to one year, after COVID-19 [126,127]. In one study, P-selectin level was increased in the hearts but not in the lungs of COVID-19 patient autopsies [4]. Crizanlizumab, a soluble P-selectin inhibitor, might increase endogenous thrombolysis in COVID-19 [122].

#### 3.2.4. Intrinsic and Extrinsic Coagulation Pathways

Coagulation pathways–inflammation–thrombosis axis could significantly contribute to the induction of COVID-19 coagulopathy. Human bronchial epithelial cells infected with SARS-CoV-2 showed upregulation of TF, the master regulator responsible for the initiation of the extrinsic coagulation pathway [128]. Cellular and humoral immune dysfunction, complement activation, IL6, and IL-1β-NETs axis also contributes to upregulation of TF and immunothrombotic events [129]. The disbalance between TF and endogenously encoded inhibitors, such as TF pathway inhibitor and protein S–protein C complex, further exacerbates the COVID-19-related thrombotic pathology [128]. Moreover, the genes for protein PAI-2 (plasminogen activator inhibitor-2), which inhibits urokinase and tissue plasminogen activator, are significantly expressed in human bronchial epithelial cells infected with SARS-CoV-2 and this contributes to pulmonary thromboses and distal coagulopathies [128]. Severe COVID-19 and pulmonary thrombosis are associated with elevated factor V and factor VIII activity [130,131]. Blood-clotting protein factor V is produced by circulating neutrophils, T-cells, and monocytes, being also expressed by lung-infiltrating leukocytes in patients with fatal SARS-CoV-2 infection [131]. The intrinsic pathway of coagulation is part of the innate immune system as an inflammatory response mechanism against SARS-CoV-2; it is also involved in immunothrombosis, being activated especially by proinflammatory cytokines such as IL-6, IL-1, and TNF-alpha [94,132].

#### 3.2.5. Fibrinolytic Disbalance and the Central Role of PAI-1

The altered fibrinolytic balance promotes the development of a hypercoagulable state that will lead to microvascular thrombosis [29]. In patients with COVID-19, impaired fibrinolysis is secondary to high plasminogen activator inhibitor 1 (PAI-1) levels, a member of the serine protease inhibitor superfamily that blocks the conversion of precursor plasminogen to active plasmin [18,133,134,135]. PAI-1 can also bind to TLR4 on macrophages and stimulates the secretion of proinflammatory cytokines and chemokines being at the crossroads of hemostatic and inflammatory pathways [136,137]. SARS-CoV-2 S protein stimulates the production of PAI-1 by pulmonary microvascular ECs using a proposed mechanism involving the disbalance in the zinc metallopeptidase STE24-ACE2 axis [133]. Inflammatory cytokines (IL-1, IL-6, IL-17A) and subsequent type II alveolar cells injury results in decreased surfactant and induction of the p53 pathway that also upregulates PAI-1 and changes the fibrinolytic balance in the lungs [138]. Higher PAI-1 levels are linked with severe COVID-19, as it is involved in EC dysfunction and lung injury by pulmonary fibrin accumulation/hyaline membrane formation and promotes local hypoxia [133,134,135,139,140,141,142]. Diffuse alveolar damage and increased hyaluronan production leads to even higher levels of PAI-1 [136].

#### 3.2.6. SARS-CoV-2–RBCs Axis

SARS-CoV-2 can infect RBCs by attaching to Band-3 protein from the erythrocyte membrane [43]. This interaction is followed by alterations in CO_2_ uptake and oxygen release from hemoglobin, inducing thrombosis by tissue hypoxia [43]. Increased plasma levels of IL-1β, IL-6, and IL-8 in patients with COVID-19 also has a negative effect on the RBCs’ ultrastructure and induces signs of eryptosis, a form of suicidal death of RBCs, that exhibits an increased tendency of adhering to ECs as well as platelets, contributing to thrombosis [93]. Moreover, COVID-19 patients’ D-dimers are correlated with RBC surface phosphatidylserine (RBC-PS), another potential contribution of RBCs in the thrombotic diathesis [143]. Induced EC dysfunction, and the presence of surface binding proteins for IL-8 and Duffy antigen receptor for chemokines on their surface [93], also link the RBCs with the immuno-stimulatory pathways and the crossroads of inflammation and thrombosis.

The main pathophysiological pathways supporting the mechanisms responsible for COVID-19-associated pulmonary immunothrombosis are summarized in Table 1.

## 4. Chest CT Imaging Data Supporting the Role of Pulmonary Immunothrombosis

Diffuse, segmental/subsegmental thrombosis is predominant in COVID-19 patients [33,34,144,145,146,147]. As small peripheral lung thrombi are more prevalent in COVID-19, evaluating the subsegmental pulmonary arteries by computed tomography pulmonary angiogram is essential [33,148,149]. In their study, Cau et al. found that in 87% of cases, pulmonary thrombosis was in lung parenchyma affected by COVID-19 pneumonia [150]. Patients with COVID-19 pneumonia and associated pulmonary thrombosis had significantly worse segmental opacifications in CT imaging and all thrombi were located in segments with inflammation [5]. The percentage of ground-glass opacities, consolidation, and crazy paving in the lobes with thrombosis was higher than in the contralateral pulmonary lobes without pulmonary thrombosis [151]. In another paper, the distribution of thrombosis correlated with the pattern of consolidation, and in 93% of cases, it involved peripheral or subsegmental arteries [152]. Patients with pulmonary artery thrombosis have significantly higher inflammatory lesions on chest CT analysis compared to those without thrombosis, especially in the lower lobes of the lungs, where blood clots predominate [5,147,153]. When analyzing quantitative chest CT data in patients with SARS-CoV-2 pneumonia and pulmonary thrombosis, more than half of them had more than 50% COVID-19-associated lung involvement, which is also associated with a higher mortality [147,154,155].

## 5. The Role of Venous Thromboembolism

In non-ICU hospitalized COVID-19 patients who underwent systematic DVT screening, when pulmonary artery occlusion appeared, this was caused by in situ pulmonary thrombosis rather than by emboli from peripheral vein thrombi [156]. In a prospective multicenter cohort study including 1240 hospitalized COVID-19 patients who performed a CT pulmonary angiography, only 11.7% of 103 patients diagnosed with pulmonary thrombosis had DVT [157]. DVT was found in 59 patients (39.3%) when Doppler ultrasound was used systematically for evaluation of the lower extremities during hospital admission for COVID-19 patients [158]. In another study, none of the patients with pulmonary thrombosis had clinical evidence of DVT [152]. More than half of patients with pulmonary thrombosis lacked signs of DVT [6,7]. Not only clinical, but also histopathological data on COVID-19 patients showed the lack of evidence for venous occlusion of the lower limbs and vessels of the pelvis [15]. Pulmonary thrombosis cases occurred in the absence of a recognizable DVT; in two-thirds of cases, most of them involving the distal vessels [159]. Peripheral pulmonary blood clots, predominant in COVID-19 as showed earlier, reflect in situ thrombosis rather than DVT embolism, which is associated with a more centrally located clot burden [149,160]. The presence of DVT at a distal level (below knee) could be an indicator for in situ thrombosis (COVID-19 thrombotic phenotype) and at a proximal level (above knee), especially if there are associate pulmonary central clots (in a main/lobar artery), could suggest an embolic origin [149]. Central pulmonary emboli are also related to lung cancer, and these patients usually have other associated risk factors for venous thromboembolic disease and worse outcomes [161]. Proximal asymptomatic DVT is also independently associated with active cancer [162]. Therefore, it is possible that some of the patients with DVT-associated pulmonary embolism could also be more prone to venous thromboembolic events. In patients with known COVID-19-associated pulmonary thrombosis, there is a lower incidence of DVT compared to non-COVID-19 patients (6.9–13.6%, vs. 45–70%), as patients with SARS-CoV-2 infection often lack traditional risk factors and comorbidities for venous thromboembolic disease [148]. Venous thromboembolism could also play a role in pulmonary artery thrombosis in some cases since COVID-19 is a known risk factor for both DVT and associated pulmonary embolism [161,163,164]. In a study on 81 ICU patients with severe COVID-19 pneumonia and lower limb venous doppler ultrasound evaluation, venous thrombotic disease was reported in 25% of cases [165]. However, considering the hypercoagulable state of COVID-19 patients and the possibility of cothrombotic events [166,167], it remains unclear if pulmonary artery thrombosis is a concurrent rather than a sequential event in patients with DVT, as specific data is missing. High rates of asymptomatic distal DVT reported in prolonged hospitalized patients with COVID-19 and no differences in prevalence of pulmonary thrombosis between patients with and without DVT could support this theory [168].

## 6. The Failure of Anticoagulation Treatment in Immunothrombosis

The failure of anticoagulation treatment in thrombotic diseases is very uncommon, but it is recognized as a problem especially among some patients with coexistent cancer [169,170,171]. The effect of anticoagulation in preventing pulmonary immunothrombosis is less known in comparison with other kinds of thrombotic events [172]. COVID-19-associated thrombotic complications could develop despite standard prophylactic anticoagulation, suggesting that a therapeutic anticoagulation or alternative antithrombotic agents could have much better results [11,19,173,174]. The failure of anticoagulation for preventing pulmonary artery thrombosis was also reported by some clinical studies on COVID-19 patients [33,147,168,175,176]. In patients with severe COVID-19, the poor response to systemic anticoagulation is supported not only by clinical studies, but also by histological examination of the lung tissue, which described a diffuse arterial thrombosis in the lungs secondary to vascular inflammatory and prothrombotic changes and hypoperfusion [177]. Moreover, the lung autopsies of COVID-19 patients also showed no differences in the prevalence of microthrombosis according to anticoagulation doses [172]. The use of intermediate instead of standard doses of prophylactic anticoagulants did not result in lower incidence of thrombosis or mortality in critically ill COVID-19 patients [178]. Pulmonary artery thrombosis and DVT reported in hospitalized patients with COVID-19 could develop despite prophylactic, intermediate, and therapeutic doses of anticoagulation [33,147,168]. In most studies, low-molecular-weight heparins (LMWHs) were used for prophylaxis of thrombosis.

Because NETs, key molecules in immunothrombosis, are positively charged, unfractionated heparin may be a more effective option for SARS-CoV-2-associated pulmonary thrombosis [50]. Heparin also replaces molecules such as histones from the chromatin backbone of NETs and destroys their stability [50]. The failure of anticoagulation in COVID-19 patients is related not only to inflammation, but also to abnormal blood flow and hypercoagulability [179,180]. Point-of-care viscoelastic methods, such as rotational thromboelastometry (ROTEM) for whole blood analysis, provide useful clinical information about clot formation and fibrinolysis, and they are useful to identify a hypercoagulable state related to severe SARS-CoV-2 infection [179]. Thrombelastography and thromboelastometry data also showed that COVID-19 patients have hypercoagulability and fibrinolysis shutdown despite the use of appropriate thromboprophylaxis [181]. According to current guidelines, therapeutic anticoagulation is now recommended for all non-ICU hospitalized COVID-19 patients with supplemental oxygen need regardless of the presence of pulmonary artery thrombosis [182]. An extensive discussion about anticoagulation strategy and other pharmacological agents is beyond the scope of this review. A combination of therapeutic strategies to prevent thrombotic events in COVID-19, beyond traditional anticoagulation alone strategies, is needed [19].

## 7. Conclusions

COVID-19-associated pulmonary thrombosis is a complex pathology, being the consequence of a complex interplay between the inflammatory response, endothelium, and coagulation systems that leads to a systemic and also local, lung-associated, procoagulant state. Immunothrombosis is the main pathophysiological result, which significantly contributes to COVID-19-associated pulmonary thrombosis; this is supported by clinical, molecular mechanisms, pathological studies, and imaging data. The high incidence of thrombotic events despite routine thromboprophylaxis in severe disease further supports this pathway. Venous thromboembolic disease could also play a role, especially in patients with other significant risk factors for thrombosis. However, considering the hypercoagulable state of COVID-19 patients and the possibility of cothrombotic events, it remains unclear if pulmonary thrombosis is a concurrent or a sequential event in patients with DVT, as specific data is scarce and requires further studies. Understanding the mechanisms behind COVID-19-induced immunothrombosis will lead to future combination therapies for hospitalized patients with severe disease that would target the crossroads of inflammatory and coagulation pathways.

## Figures and Tables

**Figure 1 biomedicines-11-00929-f001:**
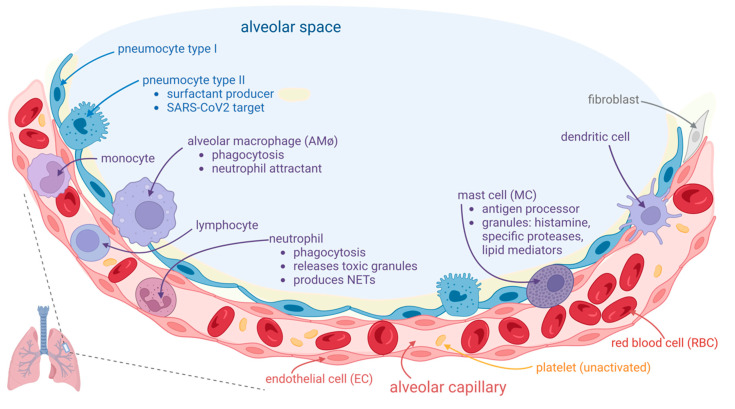
Normal host alveolar capillary associated immune defense system. The alveolar space is lined with alveolar epithelium, consisting of two kinds of cells: type I and type II pneumocyte; type II cells are involved in surfactant secretion (light yellow in the figure) and in local innate immune defense. They are also a target for SARS-CoV-2, which replicates inside these cells and can be further transferred into adjacent endothelial cells (ECs). The alveolar macrophages (AMφs) are placed into alveolar space, adjacent to the alveolar epithelium. AMφs are largely involved in immune defense, mainly by two processes: phagocytosis and chemoattraction of neutrophils. The alveolar-capillary layer is very thin over a large area, consisting of the adjacent membranes of both respiratory epithelium and vascular endothelium, and it is thicker where the body of these cells and other cells (fibroblasts in interstitium, mast cells (MCs) in subendothelial place) or other interstitial elements are placed. MCs are involved in COVID-19 pathogenicity. They contain granules with histamine and specific proteases. The capillary, lined with flat ECs, contains red blood cells (RBCs), platelets, and white blood cells. Neutrophils could pass into alveolar space if they are chemoattracted by activated AMφs. They are involved in the phagocytosis process. They contain granules with oxidants, that could be released outside the cell. They also can release extracellular traps (NETs) consisting of web-like structures containing DNA filaments coated with histones and granule proteins that can entrap and eliminate various pathogens.

**Figure 2 biomedicines-11-00929-f002:**
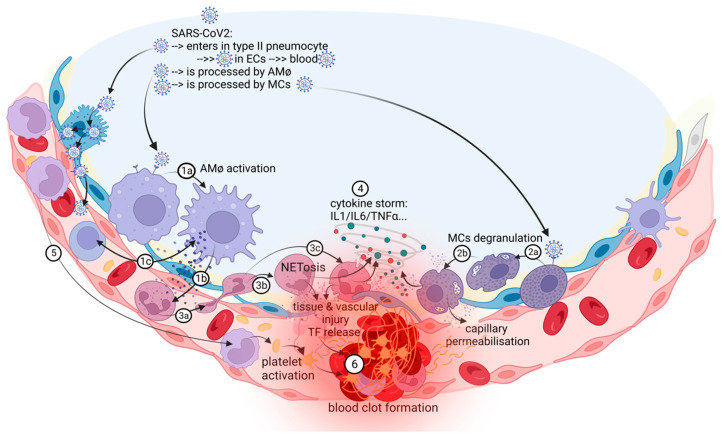
The in situ immunothrombosis in COVID-19. SARS-CoV-2 replicates into the type II pneumocytes. Furthermore, SARS-CoV-2 could be transferred to adjacent ECs that have ACE-2 receptors and passed into the blood. AMφs interact with SARS-CoV-2 and activate themselves (1a). After activation, the major process in COVID-19 is the chemoattraction of neutrophils (1b). AMφs could also act like antigen presenters to T lymphocytes (1c). Neutrophils, attracted by AMφs, come into the area of the immune conflict, crawling, and squeezing (3a), and they activate and release neutrophil extracellular traps (NETosis) (3b); aberrant NETosis and oxidants released (3c) into the environment contribute to “cytokine storm” (4), tissue and vascular injuries and initiate immune-activated coagulation (6). SARS-CoV-2 activates MCs (2a) and they release histamine, other specific proteases, prostaglandins and leukotrienes, cytokines and chemokines (2b), contributing to capillary permeabilization and to the “cytokine storm” (4), ending with in situ immunothrombosis (6). Monocytes are excessively recruited in the area (5) and contribute to the “cytokine storm” (4) and to the activation of coagulation pathways (6). The blood clot (6) contains RBCs, activated platelets, fibrin, NETs, and lymphocytes supporting the immune mechanism for thrombosis.

**Table 1 biomedicines-11-00929-t001:** Biological pathways of COVID-19 associated immunothrombosis.

Inflammation	Endotheliopathy	Coagulation
▪ Immune cells, such as macrophages, monocytes, MCs, T cells, DCs, NK cells, and neutrophils, “orchestrate” the crosstalk between proinflammatory and procoagulative networks [4,18,43,44,45,46,47,48,58,59,60,61,62,63,64,65,67,68,69,70,71,72,73,74,75,76,77,78,113,114,115,129,131]. ▪ “Cytokine storm” primes neutrophils to release NETs, which are responsible for the interplay between inflammation and thrombosis, especially in severe COVID-19 [11,18,19,45,49,50,51,52,53,54,55,56,57]. ▪ Cytokines, especially IL-1 and IL-6, and chemokines are involved in all phases of inflammation and thrombosis, using multiple biological pathways [18,29,43,44,45,50,51,58,59,60,61,62,66,69,72,73,74,75,76,77,78,79,80,81,82,83,93,94,101,103,129,132,136,137,138]. ▪ Lectin and alternative pathways mediate complement system activation by SARS-CoV-2 S and N proteins; complement system is a key mediator of the innate immune response and inflammation [11,19,84,85,86,87,88,89,90,91,92,129].	▪ Endotheliopathy is at the crossroads of hemostatic and inflammatory pathways in COVID-19 [11,18,19,93]. ▪ Cytokines, especially IL-6 and IL-1, induce and maintain microthrombi and inflammation, by promoting ECs–leukocyte adhesion and endothelial dysfunction [66,81]. ▪ In vascular ECs, IL-6 favors hypercoagulation by interfering with their normal anticoagulant and profibrinolytic properties [79,81]. ▪ ECs adopt a proadhesive phenotype in contact with SARS-CoV-2; damaged ECs lead to exposure of the prothrombogenic basal membrane and release of coagulation factors from WPBs [17,18,19]. ▪ Increased vWF expression, P-selectin, and the disbalance of sTM/nsTM expression in ECs are involved in the development of immunothrombosis, especially in severe COVID-19 [4,11,17,51,83,95,113,114,115,116,117,118,119,121,122,123,124,125,126]. ▪ SARS-CoV-2 stimulates the production of PAI-1 by pulmonary microvascular ECs, contributing to the fibrinolytic disbalance [133].	▪ Thrombocytopathy, abnormal hyper-reactivity phenotypes, and significantly elevated levels of platelet adhesion and activation mediators contribute to immunothrombi, especially in severe and critical SARS-CoV-2 infection [2,4,11,16,19,94,95,96,97,100,101,102,103,104,105,113,114,115,127]. ▪ Activated clotting factors from extrinsic, common, and intrinsic coagulation pathways are involved in SARS-CoV-2-associated immunothrombosis [11,18,19,39,47,48,66,69,72,73,77,78,81,88,89,94,128,129,130,131]. ▪ Fibrinolytic disbalance has a major role in COVID-19-associated coagulopathy, with PAI-1 as a key molecule and a marker of severe disease [18,81,94,133,134,135,138,139,140,141,142]. ▪ Functional, ultrastructure changes of RBCs and eryptosis contribute to thrombotic diathesis [11,15,16,19,43,93,143].

Legend: MCs—mast cells; DCs—dendritic cells; NK—natural killer; NETs—neutrophil extracellular traps; IL—interleukin; ECs—endothelial cells; WPBs—Weibel-Palade bodies; vWF-—von Willebrand factor; sTM—soluble thrombomodulin; nsTM—nonsoluble thrombomodulin; PAI-1— plasminogen activator inhibitor 1; RBCs—red blood cells.

## Data Availability

Not applicable.

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
