# Peer review of "Mechanisms of COVID-19 Associated Pulmonary Thrombosis: A Narrative Review"

_biomedicines, 2023, doi:10.3390/biomedicines11030929_

Round 1
Reviewer 1 Report
the manuscript is interesting, generally well written and well illustrated. Only minor comments are necessary. In particular:
Introduction: it deserves to be mentioned that SARS-CoV-2 can also lead to non restiratory diseases such as Preeclampsia, male infertility and brain damage (as recently reviewed PMID: 35114008, 35943095, 32934351). This is an important point to add since it highlights the fact that the cytokine storm found in COVID-19 patients can also damage other organs.
A table summarizing the studies discussed in the manuscript should be added.
Reviewer 2 Report
The Romanian authors summarized the current knowledge on the pathomechanisms of pulmonary thrombosis in the course of COVID-19. Although this clinical problem is widely known, the narrative review by Dr. Cristian-Mihail Niculae and colleagues is meticulous and describes in detail the individual elements leading to blood clotting disorders in COVID-19.
The paper lacks data that TEG and ROTEM provide in severe COVID-19. Authors, e.g., as the final subsection of a manuscript, may raise this issue using the following publications:
doi: 10.1055/a-1346-3178.
doi: 10.3390/life12101658.
doi: 10.3390/ijms24054319
Round 2
Reviewer 2 Report
The authors have addressed all my concerns.